# Dog–Stranger Interactions Can Facilitate Canine Incursion into Wilderness: The Role of Food Provisioning and Sociability

**DOI:** 10.3390/biology14081006

**Published:** 2025-08-06

**Authors:** Natalia Rojas-Troncoso, Valeria Gómez-Silva, Annegret Grimm-Seyfarth, Elke Schüttler

**Affiliations:** 1Cape Horn International Center (CHIC), O’Higgins 310, Puerto Williams 6350000, Chile; nrojast8@uc.cl (N.R.-T.); valeria.gomez.s@ug.uchile.cl (V.G.-S.); 2Faculty of Agronomy and Natural Systems, Pontificia Universidad Católica de Chile, Avenida Vicuña Mackenna 4860, Santiago 6904411, Chile; 3Centro Universitario Cabo de Hornos, Universidad de Magallanes, O’Higgins 310, Puerto Williams 6350000, Chile; 4Department of Conservation Biology and Social-Ecological Systems, UFZ–Helmholtz Centre for Environmental Research, Permoserstr. 15, 04318 Leipzig, Germany

**Keywords:** *Canis familiaris*, Chile, dog behavior, dog–human interaction, pet dogs, wildlife conservation

## Abstract

Free-ranging dogs are common in rural areas around the world, especially in South America. These dogs often interact with tourists, which might result in bringing them into sensitive wilderness areas, where they may disturb wildlife. In this study, we tested which stimuli (voice, touch, food) and inherent factors (age, sex, sociability) make owned and stray dogs in southern Chile follow strangers. First, we applied a 30 s socialization test, during which we stood next to the dog, recording its behavior, before offering one of the three stimuli twice for 10 s each. Thereafter, we invited the dog to follow the unknown experimenter for up to 600 m. We performed up to three trials for each of the 129 dogs and analyzed the data using multivariate modeling. We found that dogs receiving food, more sociable dogs, and dogs in the company of other dogs went for longer walks with our experimenter. Our findings highlight the importance of managing dog–stranger interactions, particularly in conservation areas. Dog owners should restrict their dogs, specifically when sociable, and tourists should be advised not to feed free-ranging dogs.

## 1. Introduction

Domestic dogs (*Canis familiaris*) are the most abundant carnivores in the world due to their long-standing relationship with humans [1]. They fulfill manifold roles, for example, as helping partners for human health and emotional support [2] or as detection dogs in biodiversity conservation [3]. Yet, dogs have recently received increasing attention as drivers of biodiversity change. In many societies, but particularly in those of the Global South, dogs are often kept unrestricted [4]; hence, they can interact with wildlife through predation, disturbance, disease transmission, competition, and hybridization (listed from most to least negative impact on vertebrates [5]). Among invasive mammalian predators, dogs were ranked number three in the severity of impact on birds, mammals, and reptiles, only behind cats and rodents [6].

Free-ranging dogs occur in diverse contexts, with varying degrees of human contact, often with interchangeable status [7]. Those dogs that are owned typically stay close to their owner’s home (e.g., in Brazil [8]; in Chile [9]; in Mexico [10]), although some individuals may roam large distances of up to 30 km (in Australia [11]) or foray overnight up to six nights (in our study area [12]). Whether dogs roam intensively depends on several factors. Better understanding the predictors of movement is crucial for wildlife conservation, as it can guide effective public awareness campaigns aimed at mitigating negative impacts. This is particularly true for husbandry-related factors; for instance, the provision of high-quality food to dogs reduces roaming [10] or the amount of wildlife they prey upon [13]. Inherent factors from the biological background of the dog cannot be modified, but understanding their effect can also help owners to evaluate the risk behavior of their dogs. It seems that male dogs are more prone to exhibit intensive forays (e.g., [14,15], but see [16]), though studies are inconsistent regarding the effect of age (e.g., [15,17]). Now, when it comes to the influence of inherent personality traits on the roaming behavior in dogs, we, unfortunately, are short of knowledge.

The study of dog personality per se is an increasing research field, although the majority of studies focus on purebred work or shelter dogs, not free-ranging dogs [18,19,20]. One dimension of personality is sociability, a trait we consider relevant for broadening the understanding of how free-ranging dogs—as highly social animals—move through the human-dominated landscape. Sociability in dogs is defined as a dog’s tendency to be friendly towards unfamiliar persons and conspecifics [21], suggested to be linked to a set of behavioral genes facilitating co-evolution with humans [22]. This trait has a genetic component and varies among different breeds (e.g., [23]), but is also influenced by environmental circumstances. For example, owned dogs showed higher levels of stranger-directed sociability when interacting on a daily basis with their owners [24]. Likewise, in a “walking away” test situation, dogs followed their owners significantly more than an unfamiliar or familiar person [25]. In the case of free-ranging dogs, which often do not have owners, Bhattacharjee et al. [26] found that urban dogs in India approached humans upon friendly cues to receive food; they even preferred to be petted by an unfamiliar person over receiving food if interaction occurred repeatedly [27]. The degree of sociability in free-ranging dogs was also shaped by the degree of human movement in urban settings [28]. However, when we consider the high variability of contexts in which free-ranging dogs occur, meaning the rural–urban, wild–domestic, socialized–unsocialized continuum, we clearly need more studies to better identify how dogs and humans are related in space.

Chile is an exceptional setting to enhance research on free-ranging dogs. Here, the overall human-to-dog ratio is as high as 4.8:1 [29], and the majority of Chile’s rural dogs are allowed to roam freely (67% in northern Chile [30]; 47% in southern Chile [31]). This creates hazardous circumstances for humans (e.g., [32]), native wildlife (e.g., [13,33]), and livestock (e.g., [34]). The literature on dogs in Chile also thoroughly documents their movement ecology through GPS-collaring (e.g., [9,12,17]) or camera-trapping (e.g., [35,36]), including the study of predictors from the human–dog interface (e.g., [15,37]). Our research was motivated by a recent study in southern Chile, which confirmed that tourists can act as vectors for free-ranging dogs in protected areas [38]. Given this, we sought to understand what type of interaction with strangers might encourage free-ranging dogs to follow them over long distances. Specifically, we aimed to answer the question: Does a friendly tone of voice, physical affection, or food influence a dog’s decision to accompany humans on long walks? Along with these stimuli, we also assessed how sociability towards humans and the presence of conspecifics might play a role in their interactions with strangers. To explore this, we exposed 129 free-ranging dogs to four experimental conditions (touch, voice, food, and control) in three cities of southern Chile where tourism plays a significant role. We expected that (1) a pronounced sociability as an inherent personality trait would motivate dogs to accompany the unfamiliar person for longer; (2) dogs would follow an unfamiliar human for longer distances when they were offered food, as food works as a reinforcer for responses in dogs [39], particularly in free-ranging dogs [40]; and (3) the repetition of trials would increase the importance of the touch stimulus on the distance a dog would follow a stranger, as prolonged social contact builds trust based on affection [27].

## 2. Materials and Methods

### 2.1. Study Area

This study was conducted in three villages in the Magallanes Region, located in the extreme south of Chile: Puerto Natales (51°43′35″ S, 72°30′22″ W), Porvenir (53°18′00″ S, 70°22′00″ W), and Puerto Williams (54°56′03″ S, 67°36′39″ W) (Figure 1). Puerto Natales is the capital of the Ultima Esperanza province, with a human population of 24,152 [41] and a human/dog ratio of 4.8:1 [42] in [43] (p. 42). The local economy is focused on livestock farming and tourism, with around 218,350 ± 60,280 annual tourists between 2012 and 2020 attracted to Torres del Paine National Park [44]. Porvenir is the capital and the largest settlement of the Tierra del Fuego province. It represents the gateway to Tierra del Fuego, with a population of 6809 inhabitants [41]; however, reliable records on tourist arrivals in Porvenir were not available. The principal sources of income are sheep farming and tourism. While free-ranging dogs are frequently reported attacking sheep (2140 predated animals between 2013 and 2024 [45]), we could not find any canine census data for this province; however, we used the equation provided by [29] and calculated a human-to-dog ratio of 5.7:1. Puerto Williams, located on Navarino Island, is the major town of the Cape Horn province, with 1750 inhabitants [41]. Free-ranging dogs were estimated at 125–160 dogs [12], that is, one dog per 11–14 inhabitants. Between 2010 and 2020, on average, 840 ± 490 annual tourists visited the southernmost trekking trails on Navarino Island (local police register in [12]). People on Navarino Island earn income from tourism, fishing, and family-run cattle farming.

During the past decade, the scientific interest in the study of free-ranging dogs in the Chilean Magallanes Region has considerably increased, which reflects the global trend to attend to a long-ignored problem. For instance, research has addressed the predictors of movement patterns of free-ranging dogs (e.g., [15,17]), their habitat use and activity patterns [12], their presence in protected areas [36], their interference with grey foxes (*Lycalopex griseus*) native to Tierra del Fuego [46], as well as the reemergence of *Echinococcus granulosus* infections, a zoonotic parasitic disease for which dogs are hosts (e.g., [47]).

### 2.2. Ethics Statement

The Ethics Committee of the University of Magallanes approved our intervention through Certificate No. 038/CEC-UMAG/2022. The owners of the participating dogs signed an informed consent form, which informed them about the project aims, its affiliation and funding, the experimental procedure to be performed, as well as the absence of risks and benefits. The owners were given a copy of the informed consent form containing the contact information of the participating scientists and the president of the Ethics Committee. If no owner was found upon several attempts asking neighbors in the dog’s vicinity, we assumed the dog was a stray dog and proceeded with the experiment without consent (30.2% of the 129 dogs). No dogs were injured during any of the trials.

### 2.3. Participating Dogs

During the tourist season (from December 2022 to March 2023), we exposed 129 free-ranging dogs to different stimuli (voice, touch, food, and control) and recorded the distance the dogs followed the experimenter (up to 600 m). To guarantee an equal selection of dogs from the three locations, we aimed at exposing at least 10 dogs per city to each stimulus (n = 42 individuals from Puerto Natales, 45 from Porvenir, and 42 from Puerto Williams). While walking through the streets, we randomly selected dogs which were apparently resident (i.e., sitting or lying in front of a house) and subsequently identified their owners (possible in 90 out of 129 dogs). To ensure our safety, we tried to consult with the owner or a neighbor whether the dog had shown aggressive behavior towards strangers and dismissed two presumably aggressive dogs. For each dog, we recorded its sex and classified its age into three categories: juvenile (from ca. six months to up to 2 years), adult (2–8 years), and senior (>8 years). If owners could not provide the age of their dog, or if stray dogs were handled, we used the dental condition for our classification, following [48]. We tried to study dogs individually, but it was impossible to prevent the approach of other dogs nearby during the experiment, so we included this condition in our statistical models (71 dogs had company during at least one trial).

### 2.4. Experimental Procedure

The experimental procedure began with the first 30 s used to establish unforced eye contact with the selected dog (hereinafter referred to as the socialization test) followed by 10 s of using friendly and welcoming expressions (e.g., “Hello dog?”, “How are you?”). The socialization test was performed with all the dogs before presenting the stimulus. After a 10 s pause, the experimenter performed the corresponding stimulus (i.e., food, touch, voice, or control). For the voice stimulus, the experimenter said welcoming words in a soft and friendly tone for 10 s using an inviting posture (example video in the Appendix A). For the touch stimulus, the researcher bent down to pet the dog, preferably on the head or back, although the location of the touch depended on the dog’s position (e.g., if the dog was lying on its back, the researcher would pet its belly) (Appendix A). For the food stimulus, a piece of sausage was placed 30 cm in front of the dog’s mouth (Appendix A). Each stimulus was repeated twice with a 10 s interval (Figure 2). After administering the stimulus, the experimenter walked away, inviting the dog to follow her, calling in a friendly manner (“come, come” or “come here”) while measuring the distance using the iPhone’s Odometer GPS application. Every 5–10 m, the experimenter repeated the calling. If the dog did not follow until reaching 30 m, the experimenter stopped the procedure; otherwise, the pace was maintained until a maximum of 600 m. The control stimulus consisted of staying next to the dog for the same time the stimulus procedure lasted, but without presenting any stimulus, and then walking away, inviting the dog in the same way as the dogs that had received a stimulus. Each test session was recorded with a GoPro camera (Model Hero 8, GoPro Inc., San Mateo, CA, USA) attached to the experimenter’s chest to record the dog’s behavior and the presence of other dogs interacting with the focus dog during the trial. To ensure consistency, the experimenter used headphones with timely oral instructions for the procedure. All the dogs were tested up to three times, with a maximum of five consecutive days during which the respective dog must have been found. After the fifth day, the experiment was considered concluded. This procedure was tested and adapted accordingly during a pilot study with three dogs, which did not participate in the final study. All the testing procedures were conducted by the same female experimenter (N.R.-T.).

### 2.5. Behavioral Analysis

To record sociability, we classified the video-recorded behavior of dogs into 14 behaviors based on the ethograms in [49], as well as the ethograms previously used in street dogs by [40,50]. The coding was facilitated by software BORIS (v7.9.4) [51]. We divided the analysis into behaviors performed during the sociability test and behaviors performed during the presentation of the stimuli. Short-duration behaviors lasting a few seconds were classified as point events, longer-lasting behaviors—as state events. After this analysis, the behaviors were classified into inactive-neutral, active-positive, and stress-related categories (Table 1).

To measure the internal consistency of the sociability-related behaviors (active-positive scale), we calculated Cronbach’s alpha, yielding a value of 0.56. Based on this result, we decided to exclude the petting-seeking behavior, which improved the consistency to 0.67. We then calculated the percentage of active-positive behaviors in relation to the total duration of the sociability test (30 s), referred to as Sociability (test). Additionally, we calculated the percentage of time spent in active-positive behaviors relative to the total time of all behaviors recorded during the 30 s of the sociability test, referred to as Sociability (overall). To see how consistent the sociability behaviors were manifested by each dog, we correlated the active-positive behaviors shown during the first 30 s with the active-positive behaviors shown after the first 30 s until the trial ended, using Spearman’s rank correlation coefficient. Finally, point events were excluded from the analysis due to their low frequency (2.9 ± 3.3 occurrences at 31 trials in 27 dogs).

### 2.6. Statistical Analysis

We used generalized linear mixed models (GLMMs) with a negative binomial distribution to predict which stimulus (voice, touch, food, or control) might influence whether and how far dogs follow strangers. The response variable was the distance the experimenter was followed over each single trial by a dog (up to 600 m). Besides the stimulus, we also included fixed effects from the dog’s background into the model (i.e., age, sex), personality traits (i.e., human-directed sociability), the company of other dogs during the procedure (i.e., company), whether or not we could find an owner, the effect of repeated procedures (i.e., days after the beginning of the first trial, number of trials), and the interaction of repeated trials and the stimulus. Finally, the ID of each dog and the location were included as random factors (Table 2). 

We used the lme4 package (v1.1-36) [52] in R Studio version 4.5.0 [53]. We tested for collinearity between numerical predictor variables with Spearman’s rank correlations and the “ggpairs” function of the GGally package (v2.2.1) [54]. As the variable Trial was strongly correlated with the variable Days (Rs = 0.93, N = 297, *p* < 0.001), we only kept Days. Similarly, we kept Sociability (test) and excluded Sociability (overall) as they were correlated (Rs = 0.91, N = 297, *p* < 0.001). Due to zero-inflated data, we chose a negative binomial error distribution with optimizer “bobyqa” and checked for overdispersion using the observed versus fitted residuals and the Pearson’s chi-squared test, yielding a non-significant ratio of 0.71 (*p* = 0.99). To select the appropriate random effect variables of the global model, we tested all possible scenarios between random intercepts of Days, Location, and ID and chose the one with the lowest AIC, which only contained the ID and Location as random intercepts. No random slopes could be tested due to boundary effects. We validated our global model using the functions “acf” and “vif” from the “car” package, and did not detect autocorrelations or further correlations among predictors, with all variables yielding gvif^(1/(2 × df)) < 5. We performed multimodel inference among all fixed-effects covariates using the “dredge” function from the MuMIn package (v1.48.4) [55], after which we calculated the importance of each variable using the summed AIC weights of each variable across all models. We performed likelihood ratio tests to obtain p-values of fixed parameters from the final model against the model without the parameter in question. Finally, we performed goodness-of-fit tests using the functions “testDispersion” and “testQuantiles” from the DHARMa package (v0.4.7) [56], yielding *p* = 0.056 and *p* = 0.005, respectively. For graphical representations, we used the packages ggplot2 (v3.5.1) [57], smplot2 (v0.2.5) [58], and ggpubr (v0.6.0) [59]. The data used for this analysis can be found in the Appendix A. 

## 3. Results

The majority of the participating dogs were male (79.1%, 102/129) and adult (61.2%, 79/129) dogs; 24.8% were classified as seniors (32/129) and 14% as juveniles (18/129). Most dogs (69.8%, 90/129) had owners who signed the informed consent form. Eight dogs (6.2%, 8/129) followed our experimenter until reaching the predefined 600 m distance, after which the experiment ended in at least one trial, while sixty-one (47.3%, 61/129) dogs did not follow the experimenter at all. On average, over all the trials, the dogs followed the experimenter for 42.5 ± 95.8 m. Under the food stimulus, the dogs were most willing to follow the experimenter (mean = 85.5 ± 133, range: 0–600 m), followed by the voice stimulus (35.7 ± 106, range: 0–568 m) and the control (31.8 ± 62.6, range: 0–238.3 m). Dogs that received the touch stimulus followed the experimenter even less than when the experimenter did not interact with the dog (16.3 ± 44.7, range: 0–212 m) (Table 3).

The best and most parsimonious model after model selection included the fixed parameters Stimulus, Company, Sociability, and Age (Table 4), with the first three being more important than the last one (Table 5), Specifically, it was the food stimulus that had the highest positive effect on following a stranger, followed by the voice stimulus as a trend. The touch stimulus, in contrast, caused an inverse reaction in the dogs, who followed the experimenter even less than in the control setting (*p*_Stimulus_ << 0.001, Table 6). The model showed that a highly sociable personality trait influenced the dogs’ decision to accompany a stranger (*p*_Sociability_ << 0.001). This trait was consistent over the overall trials, as the active-positive behaviors the dogs showed during the first 30 s of the sociability test were positively correlated with the active-positive behaviors shown after the test (Rs = 0.454, N = 294, *p* < 0.001). The dogs accompanied by other dogs were also more prone to follow our experimenter on longer walks (*p*_Company_ << 0.001). Finally, juvenile dogs also tended to follow our experimenter more often than adult or senior dogs (*p*_Age_ = 0.10) (Table 6, Figure 3).

## 4. Discussion

This research was performed in the context of biodiversity conservation. We were motivated by the fact that humans other than dog owners attract free-ranging dogs to sensitive green areas (e.g., [38,60,61]). We thus aimed at answering the question of which stimuli (voice, touch, food), in parallel with inherent factors (age, sex, sociability), motivate free-ranging dogs to follow an unfamiliar human on longer walks.

Among the three stimuli (plus control), the provision of food was the most important predictor for motivating a dog to follow a human stranger, clearly more important than the control, voice, and touch stimuli. Apparently, gently talking to the dogs had no effect as a reinforcer in our study, as it did not differ much from the control trial. Interestingly, the touch stimulus showed a negative estimate in our GLMMs, indicating that dogs in the control situation followed an unknown human even longer than dogs receiving a social reward from petting. This situation differed from what Feuerbacher & Wynne [62] found for shelter and owned dogs, namely, the preference for petting over vocal praise in concurrent choice experiments. However, the importance of food in our experiments parallels studies comparing simultaneous choice between food and petting [63]. Both owned and shelter dogs allocated more time to food than to petting when continuously delivered, but shelter dogs allocated more time to petting than owned dogs, probably because they were relatively deprived of human interaction. Owned dogs in Feuerbacher’s study clearly preferred petting from owners over strangers, as also found by Kuhne et al. [64] when measuring appeasement gestures and redirected behaviors of dogs in both situations. The touch stimulus in our study may not have worked as a reinforcer, as the majority of our participating dogs (69.8%) actually had owners and, thus, might have been less in need of social reward or even avoided physical contact with strangers due to former negative experiences. However, the comparison of two dog populations as different as pets versus unrestricted dogs with regard to their environmental and social contexts might not be valid. When we compare the results of our study to similar research on free-ranging dogs, the research performed by Bhattacharjee et al. on unsocialized, scavenging dogs in India also found a high importance of food provision by unfamiliar humans compared to social rewards, but a strong influence of positive social interactions on the attention towards human cues [65], and even a preference of petting over food when offered in long-term trials of up to two weeks [27]. This research highlights the dependence of free-ranging dogs on humans for nourishment, but also that positive human interaction can shape the behavior of free-ranging dogs.

Summarizing the aforementioned studies, food seems to be an overall reinforcer for different types of dogs to engage in human interaction, disregarding the nutritional necessity. We did not measure the nutritional status in our study to assure a non-invasive procedure, but in prior studies in our study area, mean body condition scores were not at the lower extreme of the scale from 1 (very thin) to 5 (obese) (1.9 ± 0.6 in 84 dogs in Puerto Natales [17]; 3.3 ± 0.7 in 41 dogs in Puerto Williams [15]). However, regarding the engagement with strangers, the dog’s background (i.e., the level of social deprivation), indeed, seems to matter, not at first glance, but after several trials. Thus, shelter dogs showed attachment behaviors towards a stranger after only three 10 min interactions [66]; they developed a preference for one stranger over another even more rapidly [67]. As mentioned before Bhattacharjee et al. [27] found a strong effect of preference for social rewards over food rewards following a long-term test (i.e., tests on days 0, 1, 3, 6, 10, and 15). However, against our prediction, we could not detect any effect of the repetition of trials (up to 3 over 5 days) on increasing the importance of the touch stimulus. This could be due to the fact that we predominantly worked with owned dogs, which tend to be less socially deprived (e.g., [63]), or that our trials were not intensive enough. Future research could extend the repetitions, following the scaffold offered by Nandi et al. [68], who determined that a rewarding person is recognized by free-ranging dogs in India after four interactions over four days.

Regarding the inherent factors (age, sex, sociability) motivating dogs to follow a stranger, our study revealed that juvenile dogs tended to engage with an unfamiliar human to go for a walk (see also Pérez et al. [17], who documented larger home ranges in younger dogs), possibly because boldness decreases with age [69]. Sex had no effect in our models. Human-directed sociability, however, was one of the most influential factors. This means that dogs with a higher proportion of initiating friendly behaviors towards the stranger, such as approaching, licking, playing, proximity-seeking, and tail-wagging, were more motivated to accompany an unfamiliar person than mostly passive or stressed dogs. This relationship seems intuitive. Indeed, the other way around, following an unfamiliar person has been used as a proxy to study social attraction or responsiveness to training in dogs (e.g., [70]), but mostly in pet dogs, for example, in puppies in a reduced space (3.3 m^2^) to select future companion dogs [71], or walking a dog on a leash as one aspect to test the validity of the sociability trait [72]. For free-ranging dogs, we found one study from southern Chile recording whether dogs followed a person (distance not revealed) to describe social activity in relation to surgical and chemical sterilization [73]. However, in another study from Morocco [74], no link was found between sociability and whether dogs followed an unknown experimenter. There, the researchers aimed to test whether dogs would learn from unfamiliar humans to choose the right food box by observing their choice. None of the 21 dogs followed the experimenter to the car where the experiment ended; hence, as the authors concluded, the unfamiliar humans were seen as a source of information rather than as possible partners for interaction.

Which dogs are more or less sociable depends on their genetic heredity, on the one hand, and on their socialization, on the other. Differences in sociability-related behavior among breeds (e.g., [23]) are a result of human domestication which yielded a high degree of behavioral plasticity in dogs [75]. However, social experiences also play a role. Mirkó et al. [24] showed that stranger-directed sociability was significantly higher if owners and their pets spent more time together. Definitely, it is a challenge to predict sociability in free-ranging dogs, since they are mostly mixed-breed and with difficult access to a personal history record, for example, in the so-called “community” dogs in Chile, which are dogs without an owner, but fed by the community [76]. To learn whether an unrestricted dog is sociable and thus prone to engage with human strangers, at the moment, we can only recommend observing the dog’s behavior in situ.

Whether dogs were in the company of other dogs during the performance of the trials was equally important as human-directed sociability to explain the motivation of dogs to follow our experimenter for a longer walk. This was an interesting result, as we did not include this variable a priori when designing our experiment. However, when we were confronted with the inevitable presence of other dogs approaching the scenario during our interaction with our focus dog, we decided to include this situation as a binary variable (yes or no company of other dogs) in our statistical modeling. We cannot say, though, that we evaluated conspecific-related sociability—by definition directed to unfamiliar dogs—as the accompanying dogs most probably were familiar dogs among themselves. However, we hypothesize that the presence of dogs near the focus dog and the positive influence of their company on the motivation to follow an unfamiliar person might be related to pack membership, relatively common in free-ranging dogs (the median pack size was 4 dogs based on 172 sightings in our study area [31]). The formation of packs in free-ranging dogs has been shown to relate to conflict management among free-ranging dogs (in suburban Italy [77]), communal breeding (in urban and suburban India [78]), as well as foraging associations (in urban India [79]). Though dogs differ in many aspects (genetically, morphologically, behaviorally) from their ancestor, the gray wolf (*Canis lupus* [80]), they remain cooperative carnivores to different degrees, even up to the age-graded hierarchies we know from wolves [81]. Group formation might thus play a role in diverse situations, including during interactions with unfamiliar humans. Here, we only assessed whether focus dogs were accompanied by other dogs or not, but without further knowledge of their relationship. Therefore, more research is needed that targets the interaction of dog packs with humans explicitly.

Finally, the location of the dogs did not emerge as a significant predictor in our models, but Table 3 highlights descriptive differences across the three towns, such as a lower response to the food stimulus in dogs from Porvenir, where dogs were confronted with intentional food poisoning recently (personal communication, NGO Patitas Fueginas, 30 July 2023). This highlights the importance of incorporating socioecological variables when interpreting behavioral data in human-dominated environments.

## 5. Conclusions

Our study contributed to broadening the spectrum of research on human-dog interactions, heavily biased towards pet and shelter dogs (e.g., [24,62,63]) and only recently targeted by researchers addressing the lack of information regarding free-ranging dogs, with a special focus on scavenging, unowned dogs in India (e.g., [26,27,65]). Here, we addressed human–dog interactions of well-fed free-ranging village dogs from semi-rural areas in southern Chile, mostly having owners. Against this background, we found that sociable free-ranging dogs engage with strangers on longer walks when stimulated with food and the company of other dogs. We conclude that this finding is relevant for green destinations, where tourists might positively interact with free-ranging dogs, converting themselves into vectors for the movement of dogs into wilderness (e.g., [38]). We therefore recommend that authorities take actions to raise awareness among dog owners about the risky behavior of sociable dogs and discourage tourists from feeding dogs. However, last but not least, Chile must reinforce its policies to reduce its population of free-ranging dogs and gain control over human–dog interactions so that those do not translate into the manifold evidenced negative impacts for wildlife.

## Figures and Tables

**Figure 1 biology-14-01006-f001:**
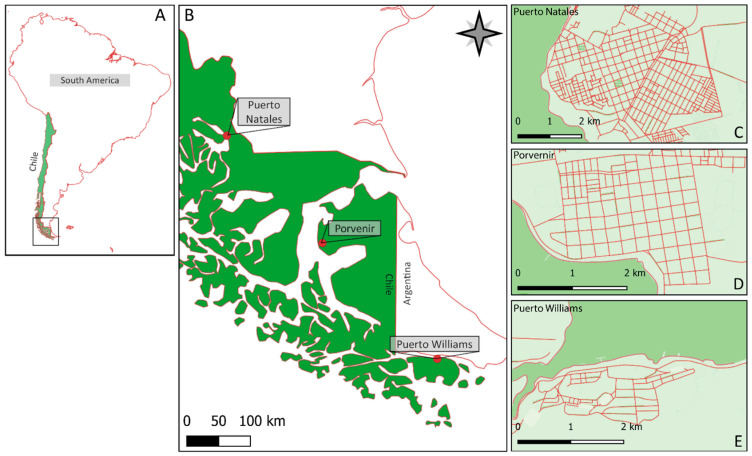
Study sites in southern Chile, South America (**A**), Magallanes Region (**B**). The behavioral experiments were performed with 129 free-ranging dogs in the cities of Puerto Natales (**C**) (n = 42), Porvenir (**D**) (n = 45), and Puerto Williams (**E**) (n = 42).

**Figure 2 biology-14-01006-f002:**
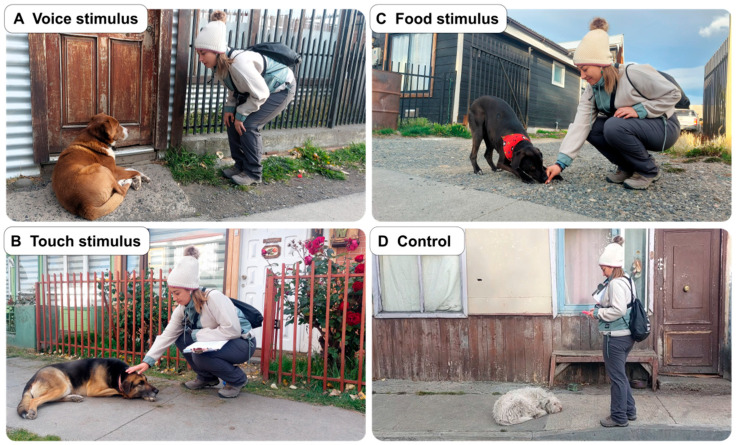
Experimental procedure for free-ranging dogs in three towns of southern Chile (Porvenir, Puerto Natales, Puerto Williams, n = 129), consisting of three stimuli and the control (same procedure, without stimulus). Each procedure consisted of (i) the socialization test (30 s), (ii) pleasant voice (“Hello dog”, “How are you?”) (10 s), (iii) break (10 s), (iv) presentation of one stimulus (10 s), i.e., (**A**) repetition of the pleasant voice (voice stimulus), (**B**) presenting a hand to dog’s nose, petting the dog (touch stimulus), (**C**) placing food on the ground, 30 cm away from the dog’s mouth (food stimulus), or (**D**) standing by the dog doing nothing (control), (v) break (10 s), (vi) repetition of the stimulus/control, and (vii) walking away, inviting the dog (“Come, come”, “Come here”), for up to 600 m. The dog owners granted permission for the use of these photos in a scientific publication.

**Figure 3 biology-14-01006-f003:**
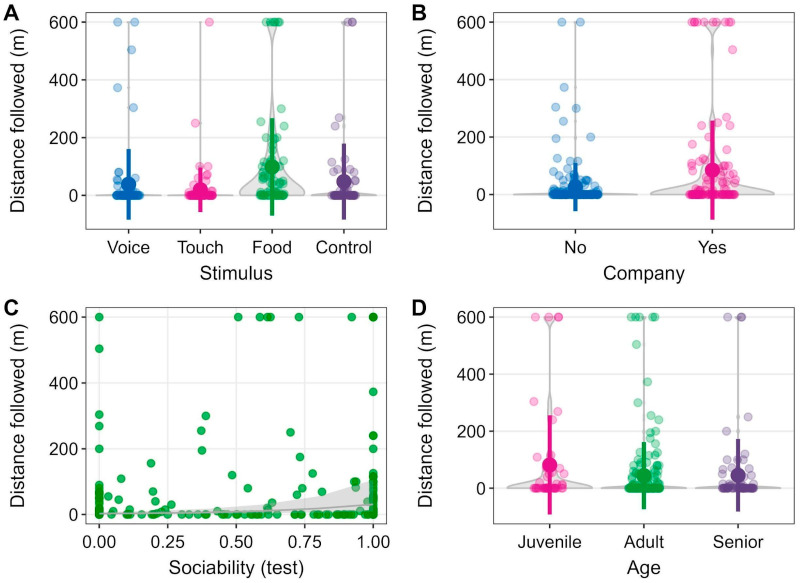
Violin charts (kernel density plots in (**A**,**B**,**D**)) and a scatterplot with the function of our model (**C**) of the most important explanatory variables in GLMMs, visualizing the influence of four stimuli (**A**), company of other dogs (**B**), sociability (**C**), and age (**D**) on following an experimenter for up to 600 m. The colored points and lines in the violin charts represent the means and standard deviations of average distances per location; the colored transparent points represent the individual data in (**A**–**D**).

**Table 1 biology-14-01006-t001:** Ethogram used to classify neutral, positive, and stress-related behaviors of the 129 free-ranging dogs performed during experimental procedures to evaluate which stimulus (voice, touch, food, and control) made dogs follow a stranger. The ethogram was based on ethograms published in [40,49,50].

Behavior	Description	Type of Event
	**Inactive-neutral**	
Sitting	The dog is sitting with a stiff neck and straightened front legs	State event
Sleeping	Lying down with eyes closed	State event
Yawning	Long drawn breath	Point event
	**Active-positive**	
Following	Active proximity and unaggressive chase	State event
Licking	Pushing tongue against others to show dominance or affiliation	Point event
Playing	Tail up, direct stare, sitting or standing alert, pricked ears, barking	State event
Proximity	Switching the position, standing, seeking petting, raising a foreleg, gazing with the tail, sniffing	State event
Seeking petting	Nosing, rubbing, or pushing contact with the person	State event
Tail-wagging	The tail moves from side to side	State event
	**Stress-related**	
Attacking	Signs of aggression that may be accompanied by chasing	Point event
Barking	Strong sound coming out of the dog’s mouth	Point event
Dominance	Tail up, direct stare, sitting or standing alert, pricked ears, barking	State event
Fear	Tail dropping, lying on the back, flattened ears, lips retracted, high-pitched whine, high-pitched bark, backing away, licking lips, retreating, scanning	State event
Laying on the back	Posture taken during play, submission, or fear (depending on context)	State event

**Table 2 biology-14-01006-t002:** Response and explanatory variables used in the generalized linear mixed models to evaluate which stimulus (voice, touch, food) and other factors predict why free-ranging dogs (n = 129) might follow strangers.

Model Variable	Description	Measurement
	**Response variable**	
Distance	Distance a dog followed the experimenter in each trial (1–3)	Continuous (0–600 m)
	**Explanatory variables**	
Age	Age class determined by the dog owner or the experimenter	Juvenile (6 months–2 years), adult (2–8 years), senior (>8 years)
Company	Other individuals present during the stimuli	Yes, no
Days	Days after the beginning of the first trial	1–5
ID	Individual dog	Continuous (1–129)
Location	Village where the experiment took place	Puerto Natales (PN), Porvenir (PV), Puerto Williams (PW)
Owner	Informed consent form signed by the dog owner	Yes, no
Sex	Sex determined by the dog owner or the experimenter	Male, female
Sociability (test)	Proportion of time of active-positive behaviors over the first 30 s	Continuous (proportion)
Sociability (overall)	Proportion of time of active-positive behaviors over all behaviors in the first 30 s	Continuous (proportion)
Stimulus	Stimulus the dog was exposed to	Voice, touch, food, control
Trial	Repetition of the stimulus during days 1–5 after the beginning of the first trial	Continuous (1–3)

**Table 3 biology-14-01006-t003:** Means and standard deviations of distances (m) over all the trials per stimulus per location in 129 free-ranging dogs in southern Chile (Puerto Natales, n = 42; Porvenir, n = 45; Puerto Williams, n = 42).

Location	Voice	Touch	Food	Control
Porvenir	37.8 ± 61.8	0.8 ± 1.7	38.6 ± 58.8	26.1 ± 37.8
Puerto Natales	9.8 ± 13.2	24.5 ± 44.4	109 ± 175	25.6 ± 54.1
Puerto Williams	57.5 ± 170	23.6 ± 63.4	116 ± 141	44.8 ± 92.3

**Table 4 biology-14-01006-t004:** Model comparison for all model combinations of the GLMMs with negative binomial error distribution within ΔAIC < 2. Selection of the best random effect structure (above) and among the fixed-effect covariates (below).

Model	df	AIC	ΔAIC
**Random effects**			
~Stimulus*Days + Sex + Age + Company + Owner + Sociability (test) + (1|Location) + (1|Individual) ^g^	17	1649.3	0.0
**Fixed effects**			
~Stimulus + Age + Company + Sociability (test) + (1|Location) + (1|Individual) *	11	1640.3	0.0
~ Stimulus + Company + Sociability (test) + (1|Location) + (1|Individual)	9	1641.1	0.76
~Stimulus + Sex + Age + Company + Sociability (test) + (1|Location) + (1|Individual)	12	1641.2	0.86
~Stimulus + Sex + Company + Sociability (test) + (1|Location) + (1|Individual)	10	1641.9	1.58
~Stimulus + Age + Days + Company + Sociability (test) + (1|Location) + (1|Individual)	12	1641.9	1.62

**Table 5 biology-14-01006-t005:** Importance values per parameter as summed AIC weights of explanatory variables across model selection.

	Company	Stimulus	Sociability (Test)	Age	Sex	Days	Owner	Days: Stimulus
Sum of weights	1	1	1	0.6	0.41	0.31	0.27	0.03
No. of models	78	93	78	79	79	94	78	32

**Table 6 biology-14-01006-t006:** Estimates, standard errors, and *p*-values of the fixed-effect variables in the final model. ^a^ Dummy coded with the control stimulus as the reference category; ^b^ dummy coded with company “No” as the reference category; ^c^ dummy coded with the age of adults as the reference category.

Parameter	Estimate	SE	*p*-Value
Intercept	0.824	0.574	
Stimulus–food ^a^	2.072	0.642	<<0.001
Stimulus–touch ^a^	−0.528	0.565	
Stimulus–voice ^a^	1.074	0.585	
Company–ye s ^b^	1.897	0.421	<<0.001
Sociability (test)	2.603	0.629	<<0.001
Age–senior ^c^	−0.560	0.519	0.096
Age–juvenile ^c^	0.859	0.546	

## Data Availability

The authors confirm that the data supporting the findings of this study are available within the Appendix A. The videos cannot be made publicly available as we granted anonymity to the dog owners through Ethics Certificate No. 038/CEC-UMAG/2022 of the University of Magallanes.

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
