# Peer review of "Dog–Stranger Interactions Can Facilitate Canine Incursion into Wilderness: The Role of Food Provisioning and Sociability"

_biology, 2025, doi:10.3390/biology14081006_

Round 1

Reviewer 1 Report

Comments and Suggestions for Authors

This is an interesting study with a well written description of the work.  The findings and conclusions are clearly presented.

There are a few wording suggestions to improve the manuscript.

Throughout the document the authors use the word "majorly".  This is a slang term and it would be better to replace that word throughout with "mostly" or "predominantly".

page 2 line 77 "the amount of wildlife in their prey"  This phrase is not clear.  Do you mean the amount of their prey that was wildlife?

page 3 line 99 "even did they prefer to be petted"  Do you mean they preferred to be petted by an unfamiliar over receiving food?

page 3 line 126 "sociability as inherent personality trait"    You are missing a word here... as an inherent personality trait.

page 11 line 383 "even more rapidly developed shelter dogs a preference for one..."  Do you mean that shelter dogs rapidly developed a preference for one stranger over another?

Some questions:

Page 4 lines 142-153:  You do not provide an estimate for the number of tourists in Provenir.  Is this because that information is not available?  If so, perhaps you can just state that.

Page 8 line 276:  "Similarly, we kept Sociability (test)...as it was correlated with Sociability (overall).."  Does this mean you used only the Sociability (test) and excluded the overall score?

page 9 lines 307-309:  I assume you do mean this is Table 3.  You have a very large difference in voice mean values, touch values, food values, and control values across the communities that might be worth mentioning in your discussion.  Do you think place matter with the experimental condition responses?  Can you provide the sample sizes by community to help the reader interpret possible explanations for findings from your models might have been influenced by location?

page 11 lines 375-280  You mention nutritional status as being unrelated to food responses based on previous research.  Is the previous work the reason you did not assess nutritional status in your study?

The figures provided in the paper are extremely helpful in visualizing the results from your work and I commend you for including them.

Author Response

RESPONSE

Manuscript ID: biology-3763399

Sociable Free-Ranging Dogs Follow Unfamiliar Humans When Receiving Food

Now entitled:
“Dog-stranger Interactions Can Facilitate Canine Incursion into Wilderness: The Role of Food Provisioning and Sociability”

Dear Mr. Martin Ma,

We would like to thank the reviewers and the editor for their careful reading and constructive feedback on our manuscript. We appreciate the opportunity to revise our work and believe that the suggested changes have substantially improved the clarity and overall quality of our study. Below, we address each of the reviewers’ comments in detail. All changes made to the manuscript have been highlighted in the revised version. The line numbers refer to the first submitted manuscript.

Yours sincerely,

Natalia Rojas-Troncoso, Valeria Gómez-Silva, Annegret-Grimm-Seyfarth, Elke Schüttler

Reviewer 1 – Comment 1
Throughout the document the authors use the word "majorly". This is a slang term and it would be better to replace that word throughout with "mostly" or "predominantly".

Response:
We appreciate the reviewer’s suggestion regarding language precision. We have revised the manuscript to replace all instances of "majorly" with the more appropriate terms "mostly" or "predominantly," depending on the context.

Line 389: This could be due to the fact that we predominantly worked with owned dogs which tend to be less socially deprived (e.g., Feuerbacher et al. 2014) or that our trials were not intensive enough.

Line 401: (...) and tail-wagging, were more motivated to accompany an unfamiliar person than mostly passive or stressed dogs.

Line 422: Definitely, it is a challenge to predict sociability in free-ranging dogs, since they are mostly mixed-bred and with difficult access to a personal history record, (...).

Line 459: Here, we addressed human-dog interactions of well-fed free-ranging village dogs from semi-rural areas in southern Chile, mostly having owners.

Reviewer 1 – Comment 2
Page 2, line 77: "the amount of wildlife in their prey". This phrase is not clear. Do you mean the amount of their prey that was wildlife?

Response:

Thank you for pointing this out. We have revised the sentence to clarify the intended meaning. The revised sentence now reads:

Line 77: "... or the amount of wildlife they prey upon (Silva-Rodríguez & Sieving 2011)".

Reviewer 1 – Comment 3
Page 3, line 99: "even did they prefer to be petted"  Do you mean they preferred to be petted by an unfamiliar over receiving food?

Response:
We appreciate the reviewer’s observation. Yes, our intended meaning was that the dogs in Bhattacharjee et al. (2017) preferred petting from an unfamiliar person over receiving food, but only after repeated interactions had occurred. To clarify this point, we have revised the sentence to read:

Line 99: "...and they even preferred to be petted by an unfamiliar person over receiving food if interaction occurred repeatedly (Bhattacharjee et al. 2017)."

Reviewer 1 – Comment 4
Page 3, line 126: "sociability as inherent personality trait". You are missing a word here... as an inherent personality trait.

Response:
Thank you for noting this. We have corrected the phrase to:

Line 126: "...a pronounced sociability as an inherent personality trait would motivate dogs...".

Reviewer 1 – Comment 5

“Page 11, line 383: “even more rapidly developed shelter dogs a preference for one…” Do you mean that shelter dogs rapidly developed a preference for one stranger over another?”

Response:

Thank you for identifying this unclear phrasing. Yes, our intended meaning was that shelter dogs developed a preference for one stranger over another more rapidly. We have revised the sentence to improve clarity as follows:

Line 383: "...even more rapidly, shelter dogs developed a preference for one stranger over another (Feuerbacher & Wynne 2017)."

Reviewer 1 – Question 1:
Page 4, lines 142-153: You do not provide an estimate for the number of tourists in Provenir. Is this because that information is not available? If so, perhaps you can just state that.

Response:

While demographic and contextual data for Porvenir were available, we were unable to locate any official records quantifying annual tourist arrivals to this locality, despite extensive efforts. To clarify this absence, we have added the following sentence to the manuscript:

Line 144-145: "However, reliable records on tourist arrivals in Porvenir were not available."

Reviewer 1 – Question 2:
Page 8, line 276:  "Similarly, we kept Sociability (test)...as it was correlated with Sociability (overall).."  Does this mean you used only the Sociability (test) and excluded the overall score?

Response:
Yes, that is correct. Due to the strong correlation between both measures (rs = 0.91), we retained only Sociability (test) in our model to avoid multicollinearity. To clarify this, we have rephrased the sentence in the manuscript as follows:

Line 276: "Similarly, we kept Sociability (test) and excluded Sociability (overall), as they were correlated (rs = 0.91, N = 297, p < 0.001)."

Reviewer 1 – Question 3:

Page 9, lines 307-309: I assume you do mean this is Table 3. You have a very large difference in voice mean values, touch values, food values, and control values across the communities that might be worth mentioning in your discussion. Do you think place matters with the experimental condition responses? Can you provide the sample sizes by community to help the reader interpret possible explanations for findings from your models might have been influenced by location?

Response:

We appreciate this thoughtful observation. We added a final paragraph to the discussion:

Page 12, line 453: “Finally, the location of the dogs did not emerge as a significant predictor in our models, but Table 3 highlights descriptive differences across the three towns, such as a lower response to the food stimulus in dogs from Porvenir, where dogs were confronted with intentional food poisoning recently (personal communication, NGO Patitas Fueginas, 30 July 2023). This highlights the importance of incorporating socio-ecological variables when interpreting behavioral data in human-dominated environments.”

We also corrected Table 5 in Table 3, thank you for noting this error.

Reviewer 1 –  Question 4:

Page 11, lines 375-280: You mention nutritional status as being unrelated to food responses based on previous research. Is the previous work the reason you did not assess nutritional status in your study?

Response:

While previous research has shown that food motivation in free-ranging dogs is not necessarily driven by nutritional need (e.g., Bhattacharjee et al. 2017), our decision not to assess body condition scores was not compatible with our experimental design as for an adequate evaluation of the nutritional status the dogs (mostly with thick furs) should have been touched. We could have assessed this variable after completing all trials, but we also believed that touching the dogs in the streets might have been considered as invasive or dangerous and possibly prevented the owner's consent for our study. To address this point transparently, we have added a clarifying sentence to the relevant paragraph in the Discussion section:

Line 379-380: "We did not measure the nutritional status in our study to assure a non-invasive procedure, but in former studies in our study area, mean body condition scores were not at the lower extreme of the 1 (= very thin) to 5 (= obese) scale.”

Reviewer 2 Report

Comments and Suggestions for Authors

Manuscript ID biology-3763399

Sociable free-ranging dogs follow unfamiliar humans when receiving food

I recommend publication of the article after minor revision.

The article describes in great detail the negative impact of free-ranging domestic dogs on wildlife. The study is well-planned and well-conducted and demonstrates how dogs that receive food and attention from tourists are more likely to follow people into nature areas.

The fact that dogs follow humans that give them food comes as no surprise. Other studies have demonstrated that (e.g. ref. 25, 59 & 69). I doubt, however, that that is the most important message that the authors want to publish. I think that a much more important message is the fact that tourists might bring free-ranging dogs into ‘sensitive wilderness where they may disturb wildlife’ (line 16-17) and I think that, if the title demonstrates this message, the article will appear more interesting to readers of the journal. I therefore suggest changing the title to something like ‘The importance of humans’ interactions with free-ranging dogs on the negative impact on wildlife’.

In addition, the conclusion of the study (to raise awareness among dog owners about the risk behavior of sociable dogs and discourage tourists from feeding dogs (line 464-465) and that Chile must reinforce its policies to reduce its populations of free-ranging dogs and gain control over human-dog interactions (line 466-467) is particularly important because it demonstrates clearly that the main problem is not free-ranging dogs but rather human behavior.

Author Response

RESPONSE

Manuscript ID: biology-3763399

Sociable Free-Ranging Dogs Follow Unfamiliar Humans When Receiving Food

Now entitled:
“Dog-stranger Interactions Can Facilitate Canine Incursion into Wilderness: The Role of Food Provisioning and Sociability”

Dear Mr. Martin Ma,

We would like to thank the reviewers and the editor for their careful reading and constructive feedback on our manuscript. We appreciate the opportunity to revise our work and believe that the suggested changes have substantially improved the clarity and overall quality of our study. Below, we address each of the reviewers’ comments in detail. All changes made to the manuscript have been highlighted in the revised version. The line numbers refer to the first submitted manuscript.

Yours sincerely,

Natalia Rojas-Troncoso, Valeria Gómez-Silva, Annegret-Grimm-Seyfarth, Elke Schüttler

Reviewer 2 - Comment 1:

The fact that dogs follow humans that give them food comes as no surprise. Other studies have demonstrated that (e.g. ref. 25, 59 & 69). I doubt, however, that that is the most important message that the authors want to publish. I think that a much more important message is the fact that tourists might bring free-ranging dogs into ‘sensitive wilderness where they may disturb wildlife’ (line 16-17) and I think that, if the title demonstrates this message, the article will appear more interesting to readers of the journal. I therefore suggest changing the title to something like ‘The importance of humans’ interactions with free-ranging dogs on the negative impact on wildlife’.

Response:

We agree with the reviewer's comment and thoroughly thought about a better title that better places our study into the context of conservation. It is now reading: Dog-stranger Interactions Can Facilitate Canine Incursion into Wilderness: The Role of Food Provisioning and Sociability”.

Reviewer 2 - Comment 2:

In addition, the conclusion of the study (to raise awareness among dog owners about the risk behavior of sociable dogs and discourage tourists from feeding dogs (line 464-465) and that Chile must reinforce its policies to reduce its populations of free-ranging dogs and gain control over human-dog interactions (line 466-467) is particularly important because it demonstrates clearly that the main problem is not free-ranging dogs but rather human behavior.

Response:

We agree that your comment is important to highlight the importance of human behavior in shaping free-ranging dogs' behaviors. Thank you.